# An Eruption of LTR Retrotransposons in the Autopolyploid Genomes of *Chrysanthemum nankingense* (Asteraceae)

**DOI:** 10.3390/plants11030315

**Published:** 2022-01-25

**Authors:** Jun He, Zhongyu Yu, Jiafu Jiang, Sumei Chen, Weimin Fang, Zhiyong Guan, Yuan Liao, Zhenxing Wang, Fadi Chen, Haibin Wang

**Affiliations:** State Key Laboratory of Crop Genetics and Germplasm Enhancement, Key Laboratory of Landscaping, Ministry of Agriculture and Rural Affairs, Key Laboratory of Biology of Ornamental Plants in East China, National Forestry and Grassland Administration, College of Horticulture, Nanjing Agricultural University, Nanjing 210095, China; 2020204041@stu.njau.edu.cn (J.H.); 2019204044@njau.edu.cn (Z.Y.); jiangjiafu@njau.edu.cn (J.J.); chensm@njau.edu.cn (S.C.); fangwm@njau.edu.cn (W.F.); guanzhy@njau.edu.cn (Z.G.); liaoyuan@njau.edu.cn (Y.L.); wangzx@njau.edu.cn (Z.W.); chenfd@njau.edu.cn (F.C.)

**Keywords:** autopolyploidization, genome duplication, evolution, *Chrysanthemum*, Asteraceae

## Abstract

Whole genome duplication, associated with the induction of widespread genetic changes, has played an important role in the evolution of many plant taxa. All extant angiosperm species have undergone at least one polyploidization event, forming either an auto- or allopolyploid organism. Compared with allopolyploidization, however, few studies have examined autopolyploidization, and few studies have focused on the response of genetic changes to autopolyploidy. In the present study, newly synthesized *C. nankingense* autotetraploids (Asteraceae) were employed to characterize the genome shock following autopolyploidization. Available evidence suggested that the genetic changes primarily involved the loss of old fragments and the gain of novel fragments, and some novel sequences were potential long terminal repeat (LTR) retrotransposons. As Ty1-*copia* and Ty3-*gypsy* elements represent the two main superfamilies of LTR retrotransposons, the dynamics of Ty1-*copia* and Ty3-*gypsy* were evaluated using RT-PCR, transcriptome sequencing, and LTR retrotransposon-based molecular marker techniques. Additionally, fluorescence in situ hybridization(FISH)results suggest that autopolyploidization might also be accompanied by perturbations of LTR retrotransposons, and emergence retrotransposon insertions might show more rapid divergence, resulting in diploid-like behaviour, potentially accelerating the evolutionary process among progenies. Our results strongly suggest a need to expand the current evolutionary framework to include a genetic dimension when seeking to understand genomic shock following autopolyploidization in Asteraceae.

## 1. Introduction

Whole genome duplication (WGD) or polyploidization is widely recognized as a significant driver of higher plant evolution [1]. All extant angiosperm species have undergone at least one polyploidization event, forming either an auto- or an allopolyploid species. In addition to polyploidization, some species long believed to be genuine diploids are actually paleopolyploids and have originated multiple times via hybridization and polyploidization throughout their evolutionary history [2,3]. *Arabidopsis suecica* [4], rice [5], and cotton [6] are all good examples of neo-polyploids. The question of why polyploids have been so successful has generated important research activity in the past decade. Significant progress towards an understanding of the polyploid genome structure and evolution in both the short-term (synthesized polyploids) and the long-term (natural polyploids) [7,8] has been made.

Polyploid species can be formed by one progenitor species (i.e., autopolyploids) or divergent parental lineages (i.e., allopolyploids, via one-step or stepwise hybridization and polyploidization) through diverse mechanisms in natural populations. In both cases, new combinations of redundant genes originate from different or similar genomes, potentially resulting in novel traits that do not exist in the contributing species, such as increasing resistance to levels of heat/cold, waterlogging/drought and salt/alkali tolerance, disease/pest resistance, and flowering time/organ size changes that might enable the offspring to thrive in harsh environments [9,10,11,12]. Additionally, redundancy may lower selective limitations on duplicated gene copies and offer new neutral sites for transcriptionally active TEs (transposable elements) to insert and accumulate until the genome is fully diploidized [13]. These changes have been detected in several diverse synthetic allopolyploids and might be associated with genomic perturbation at both the genetic and epigenetic levels. Several mechanisms that might affect these (genetic or epigenetic) alterations have occurred following polyploidization, for which the main manifestations include chromosomal rearrangement, the gain and loss of chromosome segments, gene repression and activation, and changes in the epigenome, particularly with respect to patterns of cytosine methylation [14,15]. These mechanisms might buffer the disadvantages of redundancy in the genome and might provide starting points for the evolution of new lineages under ‘‘genomic shock’’, according to McClintock [16,17,18].

Retrotransposons, as mobile genetic elements, have attracted considerable attention. These elements can move, and occasionally spread, within the host genome through the reverse transcription of an RNA intermediate [19,20]. Ty1-*copia* and Ty3-*gypsy* elements represent two major superfamilies of LTR retrotransposons in plants with similar structural features, e.g., flanking LTRs in direct orientation and multiple domains involved in different steps of element transposition [21]. Due to the repressive chromatin modifications mediated by RNA-directed DNA methylation, LTR retrotransposon activity is typically epigenetically silenced. However, under certain biotic and abiotic phenomena, host repression might fail, resulting in large-scale LTR retrotransposon activation and/or proliferation [22,23]. Recent studies have provided valuable insights into LTR retrotransposon dynamics in allopolyploids between distantly related genomes, which might be restricted to some specific elements and/or model systems. For example, the mobility of the sunfish transposon was detected in *Arabidopsis thaliana* × *Arabidopsis arenas* synthetic allotetraploids [24], where retrotransposons showed strong transcriptional activation in several hybrid sunflowers (*Helianthus* species), potentially reflecting interspecific hybridization events [25,26]. In the autopolyploidy of Buckler Mustard (*Biscutella laevigata*), the related work on LTR-RT families showed that low expression levels of transpositionally active LTR-RT families in autopolyploids further indicated that genome shock and redundancy are non-mutually exclusive triggers of LTR-RT proliferation [13]. In addition, the LTR-RT fractions in the autopolyploid genomes of *B**.laevigata* showed the similarity between the widely divergent genomes of polyploids. Remarkably, the *WIS2* retrotransposon exhibits strong transcriptional activation in wheat; however, there was no evidence of *WIS2* retrotransposon proliferation in response to allopolyploidization in synthetic allopolyploids *(Aegilops sharonensis* × *Triticum monococcum*) [27] and no evidence of mobility in response to allopolyploidization in natural *Brassica* species [28,29]. Hence, species differences in various families might affect the proportion and categories of retrotransposons. Studies have shown that genome rearrangement, the gain and loss of chromosome segments, and retrotransposon mobility might be consistent with expectations of conflict resolution in allopolyploids derived from genetically divergent parents in several species groups. However, few autopolyploids have been examined, and studies have barely focused on the responses of genome changes and retrotransposon polymorphisms in autopolyploidization. As genome shock might expectedly be minimized in young autopolyploids originating from slightly divergent progenitors, this phenomenon has limited effects on global genome structure and yields limited genome reorganization in some species [30,31]. The proportion and categories of DNA fragments affected differs in various families under polyploidization. To promote a better understanding of the success of polyploidization, further independent polyploidization events (particularly autopolyploidization) should be analyzed in future studies.

The tribe Anthemideae Cass of the Asteraceae is considered a good model system for studying the mechanisms of polyploidization. As one of the largest and most diverse families of flowering plants, Anthemideae has at least 1741 dominant species [32]. Polyploidization is a prominent driver of evolution among angiosperms [1], affecting not only the current distribution and success but also the evolution of these species. Coincidentally, polyploidization occurs with greater frequency among Asteraceae species than among other angiosperm families, and this increased frequency also provides an ideal opportunity for investigations of the rapid adaptations associated with polyploidization [33,34,35]. The Asteraceae genus *Chrysanthemum* harbours several polyploid species. Considerable variation at the ploidy level have been observed in this genus (from 2n = 2x = 18, to 2n = 36, 54, 72, up to 90) [36,37]. The publication of the reference genome of *C. nankingense* using de novo whole-genome assembly sequencing technologies allows further exploration among the *Chrysanthemum* genus [38]. Based on the reference genome data, we investigated the genomic, transcriptomic, and epigenomic alterations in allopolyploids of this genus [11,35,36,39,40]. Genome-wide transcriptomic alterations were detected in a set of synthetic allohexaploids. Rapid sequence elimination is common in *Chrysanthemum morifolium* × *Leucanthemum paludosum*, *Chrysanthemum nankingense* × *Tanacetum vulgare*, and *Chrysanthemum crassum* × *Crossostephium chinense* amphihaploids and amphidiploids [39,40]. Some polymorphic fragments of these plants were actually microsatellite sequences, suggesting that allopolyploidization might be accompanied by defects in homologous recombination and DNA damage repair mechanisms [40]. Cytosine methylation machinery responses might act as a “genome defence” system in *Chrysanthemum* spp. during allopolyploid formation, potentially reflecting a balance between the increased activity of MET1 in the higher ploidy genomes and the larger number of CpG dinucleotide sites available for methylation [35,36,40]. These findings suggest the need to expand the current evolutionary framework to encompass the genetic/epigenetic dimension when seeking to understand allopolyploidization.

In a previous study, we performed a cytogenetic analysis of autotetraploids in *C. nankingense* and observed no evidence of any chromosome elimination [40]. Although four autotetraploid lines appeared to be cytologically uniform and shared an identical phenotype, previous reports on the simple sequence repeat (SSR) analysis revealed slight polymorphic variations between these plants [40]. To a certain degree, microsatellite changes might reflect genetic variation following autopolyploidization. Considering that autopolyploidy also plays an important role in the evolution of polyploidization origins and that the rate of autopolyploid formation is higher than the number of allopolyploids, few samples were examined to characterize the response of genome changes following autopolyploidization [41]. A decade earlier, we synthesized autotetraploids of *C. nankingense* (Figure 1). Although a decade is a long genomic reaction cycle, it does not seem to be sufficient to allow a long-term process in plant evolution. Then we detected the changes in the allopolyploidization stage. Here, we used the *C. nankingense* diploid and its autotetraploids to examine rapid genome evolution following autopolyploidization using amplified fragment length polymorphism (AFLP) markers. Subsequently, we detected a Ty1-*copia* type retrotransposon with homology, suggesting that retrotransposons might activate the autopolyploidization of *C. nankingense*. In particular, we examined the characteristics and transcriptional dynamics of the reverse transcription of Ty1-*copia* and Ty3-*gypsy* elements. The activity and quiescence of individual retrotransposon subfamilies were also distinguished using high-throughput sequencing and FISH. After further amplification of the specific LTR sequences flanking the two Ty1-*copia* elements, the accumulation of Ty1-*copia* elements in the autotetraploid *C. nankingense* was also evaluated. All the findings indicated that activation and insertion of retrotransposons may occur in autopolyploidization.

## 2. Results

### 2.1. Alterations in the Genome Sequences of the Newly Synthesized Autotetraploids

A total of four individual autotetraploid *C. nankingense* lines were employed in the present study. Ten AFLP primer combinations were used to amplify 422 fragments from diploid *C. nankingense*. Obvious results revealed that autopolyploidization would have no effect on the AFLP genotype of autotetraploid compared with diploid species; however, the AFLP profiles revealed an obvious fragment variation in autotetraploid lines. For example, in autotetraploid line 1 (T1), the AFLP profiles consisted of 415 fragments, of which six novel fragments were observed, and 13 fragments present in diploid species were not detected following autopolyploidization (Figure 2). In addition, the autotetraploid lines 2/3/4 consisted of 418/415/416 fragments, of which 3/5/3 fragments were not present in diploid species, and 7/12/9 fragments were lost in the three autotetraploid lines, respectively.

To obtain additional information, the 17 novel fragments specific to the autotetraploid profile were recovered and sequenced. Among these fragments, seven yielded poor sequence qualities, and only ten fragments were successfully sequenced. We compared these ten sequences with known targets for general similarities and the presence of specific motifs in the NCBI database. The blast search revealed that only one fragment (belonging to autotetraploid (T1)) showed high similarity with a known gene (E-value = 6 × 10^−34^) as an LTR retrotransposon (Ty1-*copia* type).

### 2.2. Cloning and Analysis of the RT Region of LTR Retrotransposons

In LTR retrotransposons, the reverse transcriptase (RT) domain, encoding the enzymes responsible for the generation of a DNA copy from the genomic RNA template, is the major regulatory gene of the transposition cycle. Consequently, the replication of many transposons was first characterized through PCR amplification of the RT domain. Using two pairs of degenerated primers, the conserved fragments of the RT domains from Ty1-*copia* and Ty3-*gypsy* elements were PCR-amplified. After differential screening, sequencing, and removal of redundant sequences, a total of 76 Ty1-*copia*-RT-domain sequences (abbreviated as Ty1RT) and 50 Ty3-*gypsy*-RT-domain sequences (abbreviated as Ty3RT) were obtained from 223 positive clones.

The Ty1RT sequences ranged from 263 to 401 bp in length. Among these, sequences 263 bp in length were primarily observed, accounting for 48.7% of the observed sequence lengths, followed by sequences 266 bp in length, accounting for one sixth of the former (8.0%). The Ty3RT sequence length ranged from 421 to 433 bp, and the length polymorphism was significantly lower than Ty1RT, with a vast majority of sequences (42/50, 84.0%) 432 bp in length, without considering nucleotide polymorphisms. This analysis also revealed that all nucleotide sequences were AT-rich, with AT/GC ratios of 1.43 and 1.46 for Ty1RT and Ty3RT, respectively.

When translated into amino acid sequences using a bioinformatics approach, 40.8% (31/76) of Ty1RT and 28% (14/50) of Ty3RT sequences showed common abnormally processed transcripts, resulting in frameshifts with premature termination through the introduction of stop codons. Further analysis showed that among the 37 total 263 bp, seven 266 bp, and one 255 bp Ty1RT sequences examined, only one 263 bp sequence (2.7%) showed early termination. Without exception, all other Ty1RT sequences showed abnormal transcription (Table 1). For Ty3RT, all eight non-432 bp targets harboured one or more stop codons in the sequence, while the proportion of 432 bp sequences was 16.7% (7/42).

The remaining nucleotide sequences could be translated to amino acid sequences without any difficulties. For Ty1RT, 45 sequences had amino acid similarities of 80.1%, and all sequences contained conserved regions of the Ty1-*copia*-RT-domain, including the 5′ end of “TAFLHG”, the middle of “SLYGLKQ”, and the 3′ end of “YVDDM”. The identity of the amino acid sequences of Ty1RT-20, Ty1RT-21, Ty1RT-23, Ty1RT-35, and Ty1RT-7 was 100%, suggesting that “TAFLHGQLKETVYVSQPDGFVDPEFPNHVYKLNKALYGLKQAPRAWYDKLSSFLIANNFTKGSVDPTLFIQYHGAHILIVQIYVDDM” might be the original CnTy1RT sequence. The sequences of Ty1RT-39, Ty1RT-41, and Ty1RT-42 were also similar, indicating that these sequences were also in a primitive state in the *C. nankingense* genome (Appendix A). The sequence similarity of Ty3RT (approximately 90.0%) was higher than that of Ty1RT, and both of these sequences contained “RMCVDY” in the 5′ region and “VMPFGL” in the middle region as conserved areas of the Ty3-*gypsy*-RT domain. Although the length variation of Ty3RT was small, the amino acid variation was higher than that in Ty1RT, and only Ty3RT-14 and Ty3RT-26 were consistent at the amino acid level (Appendix A). The codon substitution and maximum likelihood models of dN (nonsynonymous) and dS (synonymous) were used for the reliable detection of positive selection at individual amino acid sites. Consequently, all Ty1RT sites showed no significant selection pressure, suggesting that these sites might maintain gene function via the reduction of nonsynonymous mutations during the independent evolutionary events of Ty1RT (Appendix A). For sites 2, 6, 72, 87, 138, 139, and 142 of Ty3RT, the positive selective effects might reflect selection pressure during evolution (Appendix A).

A WebLogo plot showed the amino acid frequencies at conserved regions within *C. nankingense*, and although most of these loci shared a highly conservative character and specificity, some sites showed similar stack heights, indicating a similar frequency of occurrence of each amino acid at that position. Comparatively fewer amino acid mutations were observed in CnTy1RT, and the stack height of each site was constant, with only limited thin stacks observed at sites 11 and 18, while in CnTy3RT thin stacks were more common at sites 14, 36, 51, 70, 113, 114, 130, 131, 142, 143, etc. (Figure 3a). However, completely different results were observed for the related Ty1RT and Ty3RT sequences from other species in NCBI. The thin stacks were more apparent in Ty1RT than in Ty3RT (Figure 3b). A phylogenetic analysis of different species using neighbor-joining trees showed that all Ty1RT sequences could be divided among four clades, while all Ty3RT sequences could be divided into five clades (Figure 3c). In the four clades of Ty1RT, the sequences from *C. nankingense* were classified into two groups (Ty1RT-42, -39, -41, -43, -44, -38, -40, and -22 vs. others). These two groups showed the most closely related similarity with the Ty1RT sequences from *Solanum lycopersicum*, *A. thaliana*, and *Prunus mume*, indicating that the Ty1-*copia* elements of these species might exist in horizontal transmission (Figure 3c). A similar pattern was also observed in Ty3RT, and all sequences could be further divided into three groups (Ty3RT-31 vs. Ty3RT-29, -5, -36, -7, -34, and -8 vs. the others) (Figure 3c). These apparent clades indicated the proliferation of corresponding LTR retrotransposons in genome differentiation. A random analysis of the sequence homology of each group showed that the sequence similarities were ~63% for Ty1RT and 79% for Ty3RT. These findings were consistent with the WebLogo plot results.

### 2.3. Transcriptional Activation of LTR Retrotransposons following Autopolyploidization

Many of the plant LTR retrotransposons are transcriptionally activated through various biotic and abiotic stress factors. Indeed, polyploidization is a strong stimulant for plants, causing genome-wide genetic perturbations, particularly in allopolyploids. In the present study, we employed diploid organisms (tissue-cultured and non-tissue-cultured) as controls to investigate the transcriptional activation of the RT domain of *C. nankingense* following autopolyploidization. The PCR amplicons were separated through polyacrylamide gel electrophoresis. The nearly invisible bands in tissue-cultured and non-tissue cultured diploid *C. nankingense* showed a complete turnaround, becoming obvious in all four individual autotetraploid plants, although LTR retrotransposon-like AFLP fragments were not detected in the autotetraploid line 2/3/4, suggesting that tissue culture could not induce the expression of Ty1RT and Ty3RT elements, while the transcriptional activation of LTR retrotransposons might result from autopolyploidization (Figure 4a).

To further verify the transpositional activity following autopolyploidization, the autotetraploid *C. nankingense* line 1 (T1), which showed fragment polymorphisms in the AFLP genotype (a fragment not present in diploid organisms) and had high similarity with known Ty1-*copia* elements in the NCBI genome database, was employed to sequence the transcriptome using high-throughput sequencing. Compared with diploid species, the autotetraploid line (T1) showed marked transcriptional regulation of approximately 60,000 detected targets, and one-third of these targets showed more than two-fold up- or downregulation at the transcriptional level (Figure 4b), involved in metabolism, cellular processes, enzyme activities, protein binding, and other processes or functions (Appendix A). Further analysis revealed that a total of 285 retrotransposon fragments were detected in the diploid or tetraploid line, including gag protein and gag-pol protein, pol-poly protein, int protein, the RT domain, and other proteins with the retrotransposon annotation. These annotations suggest the existence of retrotransposons; however, the type of retrotransposon remains unknown. Therefore, we reannotated the transcriptome sequence using NCBI genome databases and the expressed sequence tag (EST) database. These 285 retrotransposon fragments were classed as 139 Ty1-*copia*-like, 118 Ty3-*gypsy*-like, 21 non-LTR type (SINE) elements, and seven non-specific sequences (Figure 4c).

A total of 104 Ty1-*copia*-like, 92 Ty3-*gypsy*-like, and 13 non-LTR type (SINE) elements were observed in diploid lines, and most of these elements had low reads per kilobase per million reads mapped (RPKM) values, showing >100-fold lower levels than housekeeping genes, such as *EF1α* and *GAPDH*. Among the 104 Ty1-*copia*-like elements, the average RPKM value was only 1.4; however, following autopolyploidization, the average RPKM value increased to 4.6, with 49% gene activation and 25% gene silencing. In addition, 35 new Ty1-*copia*-like elements were first observed in autotetraploids. The gene activation was 3.3-fold higher than the gene silencing. A similar result was observed for the Ty3-*gypsy*-like and non-LTR type (SINE) elements, and the gene activation was three-fold higher than the gene silencing (75/25 and 12/4, respectively). The average RPKM value of the Ty3-*gypsy*-like elements in autotetraploids (T1) increased to 2.2, a value 1.6-fold higher than that observed in diploid species (Figure 4c).

### 2.4. Distribution of LTR-RTs in the Diploid and Tetraploid of C. nankingense

To further determine the LTR-RT sites of Ty1-*copia* and Ty3-*gypsy*, the diploid and tetraploid line 1 (T1) of *C. nankingense* were analyzed using FISH. The hybridization probes were labelled by the plasmid containing the sequence of Ty1-*copia*-RT and Ty3-*gypsy*-RT respectively. The FISH analyses of somatic metaphase chromosomes showed differential LTR-RT distribution patterns for Ty1-*copia* and Ty3-*gypsy*. As shown in Figure 5, the probe of Ty1-*copia*-RT and Ty3-*gypsy*-RT merged the different dense signals from the chromosomes. From the FISH result of the Ty1-*copia*-RT probe, we examined the spot signals on the pericentromere regions and minor dispersed signals observed throughout the chromosomes of the diploid of *C. nankingense*. However, the diffuse signals of Ty1-*copia*-RT were detected throughout the chromosomes of the tetraploid. Additionally, the high intensity signals of Ty3-*gypsy*-RT were detected on the chromosomes in the tetraploid of *C. nankingense*. In general, the FISH signals of Ty1-*copia*-RT and Ty3-*gypsy*-RT increased during autopolyploidization (Figure 5a,c). Compared with the signals of Ty3-*gypsy*-RT, the increasing signals of Ty1-*copia*-RT especially showed in the merged image of all channels (Figure 5b). Furthermore, we counted five cells to obtain the ratio of Ty1/Ty3, the mean value of fluorescence intensity measured by Image J increasing by 10.72% after the autopolyploidization of *C. nankingense* (Figure 5c).

### 2.5. LTR Region Analysis of Ty1-copia-like Elements

Based on the transcriptome database, we selected three Ty1-*copia*-like elements with higher upregulation expression alterations and high homology with known retrotransposons in the autotetraploid (T1) as targets for cloning and sequence analysis. Unfortunately, only two of these targets were successfully amplified, reflecting the high polymorphism of this subtype of retrotransposon. The 4598 and 4616 bp sequences were renamed CnMp1 and CnMp2, respectively.

Sequence analysis revealed that the CnMp1 and CnMp2 sequences contained conserved polyA sites in the LTR region. The boundary sequence (CA) of R-U5 was also detected among the 40 amino acid sequences of the LTR region. The PBS region also harboured the characteristic base sequence of “TGGT”, followed closely by LTR regions at a distance of 3 and 0 bp, respectively. For the coding regions, the GAG-PR-INT-RT-RNaseH regions were closely interlinked, and the GAG region contained the conserved nucleic acid-binding sequence “C-C-H-C”. The PR region followed, and CnMp1 contained the conserved “D (T/S) G” sequence, while the last amino acid of CnMp2 was “A”, which was not conserved. The INT regions of these two sequences had a conserved zinc finger domain (H-H-C-C). Three motifs, “AFLHG”, “YGLKQ”, and “YVDDM”, in the RT region and two motifs, “KHID” and “KHID”, in the RNaseH region were all relatively conserved (Figure 6). The analysis revealed that the cloned sequence was a Ty1-*copia*-type retrotransposon.

### 2.6. SSAP Analysis of the Insertion Sites in CnMp1 and CnMp2

The LTR region was sequence-specific for a given retrotransposon. The insertion of a retrotransposon into the plant genome, as opposed to DNA transposons, did not result in sequence divergences; rather, stable insertions in the genome that could be exploited using recently developed marker systems were shown. The retrotransposon-based sequence-specific amplification polymorphism (SSAP) is a traditional molecular marker technique based on insertion polymorphisms generated through the specificity of an oligonucleotide primer anchored on the terminal sequences of a retrotransposon, typically in the LTR region [42].

In the present study, the primers were derived from the LTR regions of CnMp1 and CnMp2, reflecting the actual loci of CnMp1 and CnMp2 in the autotetraploid (T1) genome. A total of fourteen primer combinations (seven combinations for each element) produced a multi-fragment SSAP profile. Most of the fragments of diploid species were inherited by the autotetraploid (T1) line; however, five of the CnMp1 primer combinations generated at least one polymorphic fragment, while only two CnMp2 primer combinations showed certain polymorphisms, generating up to 12 polymorphic fragments, suggesting the activation and insertion of the retrotransposon following autopolyploidization (Figure 7a,b). Notably, we found a similar transcript abundance of CnMp1 and CnMp2 in the autotetraploid (T1) (RPKM 6.9 vs. 7.6); however, CnMp2 was also transcribed in the diploid organism (RPKM = 3.0). Moreover, the total and polymorphic numbers of SSAP fragments of CnMp1 were higher than those of CnMp2, suggesting that CnMp1 might be a relatively young retrotransposon and/or that a mechanism might exist in *C. nankingense* for the inhibition of the CnMp2 insertion—a possibility which requires further testing.

## 3. Discussion

### 3.1. Effect on the Genome Sequence following Autopolyploidization

Polyploidization has been considered a pervasive force in plant evolution. The available data indicate considerable genomic rearrangements, including exchanges between genomes and gene loss, following allpolyploidization [14]. In the early stages of the artificial synthesis of allopolyploids between wheat and its near relatives, sequence elimination is a major immediate response of the wheat genome to hybridization or allopolyploidization [43]. Similar measurable degrees of sequence elimination were also generated in cucumber [44], cabbage [45], and salsify [46]. Although remaining elusive and receiving little attention compared with allopolyploidization, the limited currently available evidence suggests that autopolyploidization might also effect genome structure via a series of genome-wide genomic perturbations, which might rapidly change and play an influential role in genome reorganization. In the present study, the *C. nankingense* autotetraploid showed 1.6–3.0% fragment elimination and 0.7–1.4% novel fragment emergence, which may lead to convergent genome sizes to a greater extent. Although this finding was inconspicuous and not larger than the scope of *Chrysanthemum* allopolyploidization in our previous investigation [11,39], the results also reflect rapid genetic changes in the *C. nankingense* autotetraploid to a certain extent.

Compared with the predominant formation of bivalents in allopolyploids (AABB), autopolyploids (AAAA) might easily show multivalent formation during meiosis. Clausen et al. [47] and Stebbins [48] proposed that abnormal meiosis (e.g., multivalent formation) and reduced fertility compared with the diploid progenitor in autopolyploids might represent an evolutionary disadvantage. Instead, the successful range expansion and radiation demonstrated in various natural autopolyploids were much more common than traditionally anticipated, suggesting genome multiplication [49]. The benefit was obvious for autopolyploidizaton; thus, we hypothesized that the rapid genetic changes might provide a short-term evolutionary channel to increase the genome flexibility of autopolyploids via the accumulation of minor differences in the chromosome during early generation, and the cumulative effects of variations in genetic redundancy (potential reversion to a diploid-like status) might represent an evolutionary advantage on a longer evolutionary timescale.

### 3.2. Genetic Heterogeneity of the LTR Retrotransposon

The phenomenon of retrotransposon activation has been widely observed in plant evolution. In the present study, we cloned the RT domains of Ty1-*copia* and Ty3-*gypsy* elements. The transposition of retrotransposons in most eukaryotic organisms requires RNA intermediates via DNA–RNA–DNA models [50]. In this process, retrotransposons must encode reverse transcriptases and replicate through RNA intermediates, providing a key to identify the evolutionary relationships among different retrotransposons, as RTs do not have rigorous proofreading functions (substitution frequencies of approximately 10–4 per base) and could be influenced through many factors, including mutations, retrotransposition cycles, pressure selection, differentiation, population size, competition, etc., resulting in high base mutations, deletions, insertions, substitutions, frameshifts, and genetic rearrangements, particularly in the RT domain [51]. Among the 121 Ty1RT clones and 102 Ty3RT clones obtained in the present study, the length of the Ty1RT sequence ranged from 216–401 bp, where 48.7% of sequences were 263 bp in length, and the length of the Ty3RT sequence ranged from 421–433 bp, where 84.0% of the sequences were 432 bp in length, showing a large sequence length variation of Ty1RT compared with Ty3RT.

In addition, 40.8% of Ty1RT and 28% of Ty3RT sequences showed premature termination, indicating that early termination explained the reverse transcription inhibition. In addition, the likelihood of the premature termination of the 263 bp Ty1RT sequence was low (2.7%), and all seven 266 bp Ty1RT sequences and only one 255 bp Ty1RT sequence were observed, while the other Ty1RT sequences harboured one or more stop codons (Table 1). Similarly, all non-432 bp Ty3RT sequences showed abnormal transcription; however, the probability of the premature termination of the 432 bp sequence was 16.7%, 6.4-fold higher than that observed for the 263 bp Ty1RT sequences. The lower length variation (51.3% non-263 bp Ty1RT vs. 16.0% non-432 bp Ty3RT) and high base mutation in the Ty3RT sequence indicated that mutations contribute to Ty3RT premature termination. Based on the maximum likelihood codon neutrality analysis of Ty3RT, a total of seven codons showed evolutionary selection, i.e., Ty3RT might suffer certain selection pressures and undergo mutations during the evolutionary process [42], resulting in premature termination, and, among these basic mutations, sequence length variations might directly reflect the heterogeneous nature of the two retrotransposon types. Previous studies also showed that most retrotransposons are non-autonomous (i.e., their domains are absent or non-functional) [52]. This may be tendentious for autopolyploidization, as element inactivation is generally favourable for the host. Retrotransposons are more ancient transposable elements, widely distributed throughout the plant kingdom, with a high degree of heterogeneity [50]. As the prevalent component of most plant genomes, these elements can be passed on from parent to offspring (vertical transmission) and be transferred between different species (transverse transfer) [53]. Cluster analysis revealed that either Ty1RT or Ty3RT series harbour a certain similarity with other species, suggesting the lateral transfer of retrotransposons with a common origin among these species. Comparatively, the amino acid polymorphisms in Ty1RT were fewer than those in Ty3RT in *C. nankingense*; however, within different species, Ty3RT was more conserved than Ty1RT. This considerable variation of Ty1RT in different species might reflect sequence diversification after original species differentiation, while for Ty3RT, the amino acid variations might not be as dynamic as that of Ty1RT during *C. nankingense* evolution, resulting in populations of highly similar sequences of quiescent sequence lengths, particularly under a certain selection pressure.

### 3.3. Transcriptional Activity of Retrotransposons

Retrotransposons significantly affect the genetic stability of the plant genome, especially regarding the large-scale structure of complex genomes [19]. Although plant retrotransposons are silenced in a vast majority of cases, retrotransposons could be transcriptionally activated through various factors, such as numerous biotic and abiotic stresses. Polyploidization is an extremely strong shock and can activate quiescent transposons/retrotransposons, except for the mutations that have occurred in the studies of some allopolyploid species [54]. For example, the *copia*-like wheat retrotransposon *WIS*2-1A can be induced through allotetraploidization, affecting the expression of adjacent genes [27]. New insertion sites of retrotransposon *TNT*1 were detected in synthetic allotetraploid tobacco [55]. Long fragment genome sequencing analyses of ancient maize polyploid genomes showed that the activation and amplification of LTR-retrotransposons resulted in an explosive increase in maize genome size from approximately 1200 to 2400 Mb in the past three million years [56], while in cotton and the allotetraploid *Triticum dicoccoides*, a high level of intergenomic invasion of retrotransposons in the early stages of polyploid formation might reflect general tendencies in speciation and the stabilization of the allopolyploid genome [57]. In the present study, polymorphic AFLP fragments matched the Ty1-*copia*-type retrotransposon sequence, suggesting that autopolyploidization might also accompany the perturbations of retrotransposons.

Using RT-PCR and transcriptome sequencing, retrotransposon activity in the *C. nankingense* autotetraploid was examined in the present study. All detected retrotransposons could be divided into three categories and classed according to the NCBI genome database, primarily showing Ty1-*copia*-like elements, followed by Ty3-*gypsy*-like elements and several non-LTR type (lines) elements. Retrotransposons are typically silenced; however, studies have shown the transcription of a large proportion of retrotransposons in the human genome, of which at least 120 retrotransposon elements have functional implications or have evolved into bona fide genes [58]. As an indicator of functionality, we used the ratio of nonsynonymous substitutions to synonymous substitutions per site (dN/dS) to evaluate positive selection at individual amino acid sites. We observed that the Ty1RT sites had undergone no significant selection pressure, suggesting that these sites were predominantly shaped through purifying selection throughout evolution, representing potential transcriptional activity and the evolution of functional genes under selective constraints. In the present study, although with a low expressive abundance, some transcriptional retrotransposons were observed in diploid lines, whether these elements profit from the genomic environment for transcription and functionality without generating transcriptional “noise” warrants further study. More importantly, following autopolyploidization, the variations in the transcriptome were detected based on the analysis of the differentially expressed genes of diploid and tetraploid *C. nankingense* (Figure 3b). Furthermore, the retrotransposon activation (approximately three-fold higher than the silenced condition) and increases in RPKM in Ty1-*copia*-like or Ty3-*gypsy*-like elements and the number and proportion of Ty1-*copia*-like elements were higher than those of the Ty3-*gypsy* element. In addition, we found that the ratio of Ty1-*copia*/Ty3-*gypsy* elements was increased after the autopolyploidization of *C. nankingense* (Figure 5c). The result of FISH indicated that autopolyploidization might also be accompanied by perturbations of LTR retrotransposons, and the emergent retrotransposon insertions might show more rapid divergence. Therefore, retrotransposons might participate in the genomic evolution of the autotetraploid *C. nankingense*, thereby confirming initial hypotheses.

### 3.4. Dynamics of Retrotransposons following Autopolyploidization

Retrotransposons are reverse transcribed through a self-encoded reverse transcriptase, and the resulting DNA is integrated into the genome. This “copy and paste” mode of transposition increases the copy number of these elements, likely providing direct evidence of retrotransposon-mediated variations in genome size within the plant genome [19]. After about ten years of natural evolution, we conducted a FISH analysis to visualize the variations of Ty1-*copia* and Ty3-*gypsy* during the allopolyploidization of *C. nankingense*. In addition, the SSAP results suggested that activated CnMp1 and CnMp2 were inserted into the new loci of the autotetraploid *C. nankingense* genome, indicating that autopolyploidization leads to retrotransposon activation and insertion (Figure 7). It has recently been suggested that retrotransposons influence genome evolution through a high degree of adaptive or evolutionary potential in angiosperms, resulting in an extraordinary array of genetic changes, including gene modifications, duplications, altered expression patterns, and exaptation to create novel genes, with occasional gene disruptions [19,59]. Compared with allopolyploids, the bottleneck of abnormal meiosis for autopolyploids was amplification, and, to restore gamete fertility, the emergence of retrotransposon insertion might show faster divergence, resulting in diploid-like meiotic behaviour realized in limited generation, whereas the evolutionary potential of a retrotransposon insertion can extend for thousands or millions of years [19].

Asteraceae have high frequencies of hybridization and polyploidization (often at the same time), and the offspring of hybridization and polyploidization frequently show retrotransposon activation–insertion and beneficial character variations [32]. In *Achillea borealis* (Asteraceae), following polyploidization, tetraploid species showed ecological tolerance and occupied new habitats previously unassailable by diploid species [49]. Among three diploid sunflower (*Helianthus*) hybrids, retrotransposons were enhanced and inserted into the offspring genome following hybridization, resulting in a 50% increase in drought and salt tolerance compared with either diploid parent [25,26]. We previously observed higher resistance to cold, drought, and salt stress in the autotetraploid *C. nankingense* compared with diploid lines [60], while in the present study the SSAP profiles supported dramatic retrotransposon insertions in the genomic evolution of the autotetraploid *C. nankingense*. As retrotransposons are abundant and dispersed throughout genomes, these elements might result in the rapid shuffling of newly formed Asteraceae polyploid species via the contribution of abundant raw material at a critical moment in adaptive evolution, ultimately leading to the spectacular success of Asteraceae. Accordingly, future studies addressing the evolutionary consequences of polyploidization should focus on the specific evolutionary impact of retrotransposons.

## 4. Materials and Methods

### 4.1. Plant Material and DNA Extraction

The diploid (2n = 2x = 18) and four autotetraploid (2n = 4x = 36) *C. nankingense* lines were maintained at the *Chrysanthemum* Germplasm Resource Preserving Centre, Nanjing Agricultural University, China (32°05′N, 118°8′E, 58 m elevation). All investigated plant materials were identified using flow cytometry, cytogenetics, and morphology. All plants were propagated ten years through cuttings, and the medium contained a 2:2:1 (*v*/*v*) mixture of perlite, vermiculite, and leaf mould, respectively. Rooted seedlings were grown in a greenhouse at 22 °C during the day and at least 15 °C during the night, with a relative humidity of 70–75% under natural light. The experiments included three biological replications.

The genomic DNA of diploid and autotetraploid *C. nankingense* species was extracted from young leaves using a cetyltrimethyl ammonium bromide (CTAB)-based method [61]. The DNA was treated with RNase to remove any RNA contamination. The concentration and purity of the DNA preparations were measured using a NanoDrop HND-1000 spectrophotometer (Nanodrop Technologies, Wilmington, DE, USA), and the OD260/OD280 ratio was calculated.

### 4.2. Flow Cytometry

All samples were placed in 500 uL Nuclei Extraction buffer, chopped with a sharp blade, then filtered through a 50 um filter after 60 s, followed by the addition of 2000 uL of DAPI staining buffer for 2 min in dark conditions. Nuclei suspensions were analyzed with a CyFlow SpaceFlow Cytometer (Sysmex Partec, Muenster, Germany) and the corresponding FloMax software. *C. lavandulifolium* was identified as the internal reference marker plant in the flow cytometry works.

### 4.3. AFLP Fingerprinting

The DNAs were subjected to AFLP profiling according to Vos et al. [62]. Approximately 500 ng of DNA was digested overnight at 37 °C with 10 U each of *EcoR*I and *Mse*I (New England Biolabs, Beijing, China). Subsequently, adaptor sequences were ligated onto the ends of the digested fragments at 16 °C for 4 h, and the ligation mix contained the digested fragments, 5 pmol *EcoR*I, 50 pmol *Mse*I adaptors (Appendix A), and 4 U T4 DNA polymerase (NEB). The enzymes were subsequently inactivated at 70 °C for 15 min. The amplicons generated from this template based on *EcoR*I and *Mse*I primers lacking any selective bases (Appendix A) were diluted 1:30 in ddH_2_O and were used as templates for subsequent selective amplifications based on *EcoR*I and *Mse*I primers carrying three selective bases, totalling ten primer combinations of *EcoR*I selective primer #2 plus *Mse*I selective primer #5 (abbreviated “E2 + M5”), E2 + M6, E3 + M2, E4 + M3, E4 + M8, E5 + M4, E6 + M7, E7 + M3, E8 + M3, and E8 + M7 (Appendix A). The *EcoR*I primers were labelled with 5-FAM. Each PCR and electrophoretic separation was repeated twice. The amplicons were separated using an ABI 377 DNA Analyzer, and only reproducible fragments ranging from 80 to 450 bp of two replications were recorded as present (1) or absent (0).

### 4.4. Isolation and Sequencing of Informative AFLP Amplicons

Informative AFLP amplicons were also separated through electrophoresis on 6% denaturing polyacrylamide gels, followed by silver staining [11,62]. Autotetraploid-specific AFLP fragments were cut from the gel and eluted in 50 μL of ddH_2_O after boiling for 10 min. After centrifugation (12,000× *g*, 5 min), an aliquot of the supernatant was collected and served as the template. The fragment was reamplified using the same primer combination and PCR conditions described above. The resulting amplicon was inserted into the cloning vector pMD19 TA (Takara, Beijing, China) and sequenced. The similarity of these sequences to those in public databases was obtained using BLAST analysis (http://www.ncbi.nlm.nih.gov/BLAST/ (accessed on 1 October 2021).

### 4.5. Isolation of the RT Region and Transcriptional Analysis of the LTR Retrotransposon

Total RNA from the young leaves of diploid and autotetraploid *C. nankingense* lines was isolated using the Total RNA Isolation System (Takara, Beijing, China), according to the manufacturer’s instructions. Prior to reverse transcription, 30 ng of RNA was treated with 10 U of RNase-free DNAase I (Takara, Beijing, China) at 37 °C for 30 min to remove any contaminating genomic DNA. A 1 μL aliquot of the resulting RNA (containing approximately 600 ng) was used as the template for the synthesis of first-strand cDNA using SuperScriptIII Reverse Transcriptase (Takara) primed with random hexamer primers. The subsequent PCR involved two degenerate primer pairs targeting the RT regions of Ty1-*copia* (F: ACNGCNTTYYTNCAYGG, R: ARCATRTCRTCNACRTA) [63] and Ty3-*gypsy* (F: AGMGRATGTGYGTSGAYTAT, R: CAMACCMRAAMWCACAMTT) [64] retrotransposons. Subsequent phylogenetic (homologue sequences of other species were the most closely related sequences in the NCBI database according to the BlastX results) and dN (nonsynonymous)/dS (synonymous) analyses were performed using MEGA 5.0 software (http://www.megasoftware.net/mega.php (accessed on 1 October 2021) by means of neighbor-joining trees. Diploid DNA was used as a positive control, and the transcriptional profiling of the RT regions of Ty1-*copia* and Ty3-*gypsy* retrotransposons in diploids (tissue-cultured and non-tissue-cultured) and autotetraploids was performed using denaturing polyacrylamide gels.

Total RNA from the young leaf of diploid and selected autotetraploid *C. nankingense* lines was isolated using the Total RNA Isolation System (Takara, Beijing, China), according to the manufacturer’s instructions. The RNA quality (RNA integrity number (RIN) > 8.5 and 28S:18S > 1.5) was verified using a 2100 Bioanalyser RNA Nanochip (Agilent Technologies, Santa Clara, CA, USA), and the concentration was determined using an ND-1000 Spectrophotometer (NanoDrop, Wilmington, DE, USA). The applied standards were 1.8 < OD260/280 < 2.2 and OD260/230 > 1.8. Illumina (San Diego, CA, USA) sequencing was performed using an Illumina HiSeqTM 2000 device at the Beijing Genomics Institute (Shenzhen, China; http://www.genomics.cn/index.php (accessed on 1 October 2021), according to the manufacturer’s protocols. Transcriptome assembly was achieved using the short-read assembly program Trinity [65]. Functional annotation was assigned using the protein (Nr and Swiss-Prot), Clusters of Orthologous Groups (COG) and Gene Ontology (GO) databases. BLASTX was employed to identify related sequences in the protein databases based on E-values of less than 10–5 [66]. In addition, all transcriptome sequences were reannotated using the NCBI genome databases or the EST database.

### 4.6. Cytological Preparation, Probe Labelling, and Image Processing

Chromosome spread preparation and FISH were performed as previously described [67]. The plasmid contained Ty1-*copia*-RT and Ty3-*gypsy*-RT were labeled with Alexa Fluor-488-5-dUTP (Thermo Fisher Scientific, Waltham, MA, USA, Category Number C11397) and Alexa Fluor-594-14-dUTP, partly using a nick translation. The FISH images were acquired using an epifluorescence Olympus BX61 microscope (Olympus China Inc, Beijing, China) with the same exposure time and were processed with Adobe Photoshop 2021. The mean intensity of fluorescence signals was quantified using ImageJ.

### 4.7. Isolation of LTR Regions and SSAP Fingerprinting

Three Ty1-*copia*-like elements with higher upregulated expression alterations and high homology with the known retrotransposons in the autotetraploid (T1) were selected as targets to clone and analyze the sequence using thermal asymmetric interlaced PCR (TAIL-PCR) [68] and ligation-mediated PCR (LM-PCR) methods [69]. The SSAP protocol was similar to the AFLP procedure of Vos et al. [62], except for the LTR-specific primers (LSPs) used in the second amplification, which were designed using Primer 5.0. The sequences for CnMp1 and CnMp2 were 5′-TAGACCAAAATTACTAAACTACCC-3′ (LSP1) and 5′-SSAGTCTTTTTCTTTCTCATTTCTTCTC-3′ (LSP2), respectively. In the present study, LSP1 and LSP2 were labelled with 5-FAM and combined with the standard AFLP *Mse*I selective primers (M2, M3, M4, M5, M6, M7, and M8), respectively. Each PCR and electrophoretic separation was repeated twice. The amplicons were separated using an ABI 377 DNA Analyzer, and only fragments ranging from 80 to 450 bp were considered.

## Figures and Tables

**Figure 1 plants-11-00315-f001:**
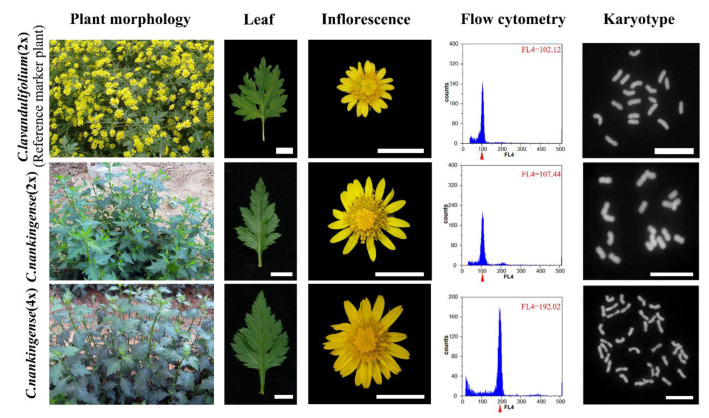
Identification of materials using flow cytometry, cytogenetics (bar = 10 μm), and morphology (bar = 1 cm). (The reference peak of diploid materials is 100 in flow cytometry examination).

**Figure 2 plants-11-00315-f002:**
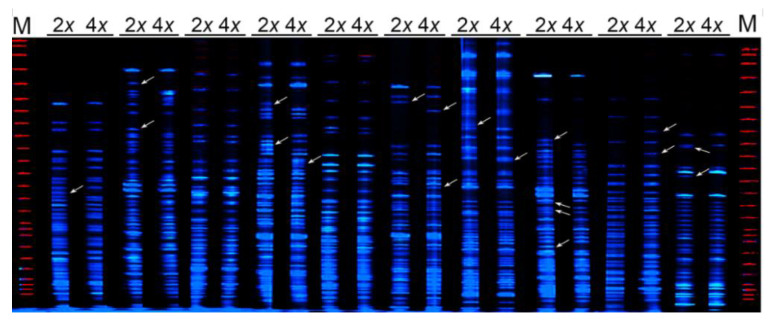
DNA-AFLP profiling of diploid and autotetraploid line 1 (T1) of *C. nankingense*. The arrows indicate genome changes.

**Figure 3 plants-11-00315-f003:**
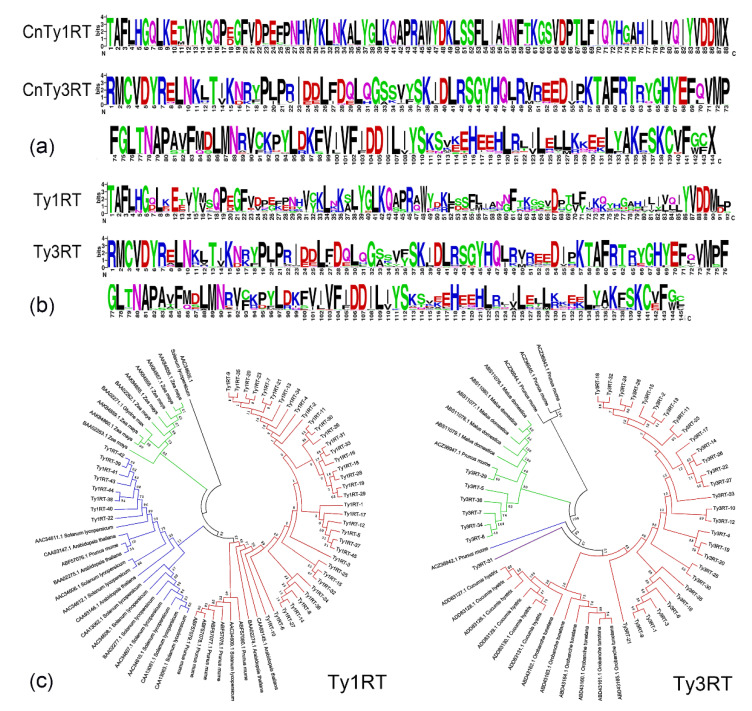
Amino acid analysis of the RT region of the LTR retrotransposon. (**a**) WebLogo plot of amino acid frequencies at conserved regions of Ty1RT and Ty3RT of *C. nankingense*. Positions are relative to the 5′ end of the target site. N, length of the target site. Logos were generated using WebLogo. (**b**) WebLogo plot of amino acid frequencies at conserved regions of the Ty1RT and Ty3RT sequences of different species. Positions are relative to the 5′ end of the target site. N, length of the target site. Logos were generated using WebLogo. (**c**) The phylogenies of Ty1RT and Ty3RT using neighbor-joining trees. According to the BlastX results, homologous sequences from other species were the most related sequences in the NCBI database.

**Figure 4 plants-11-00315-f004:**
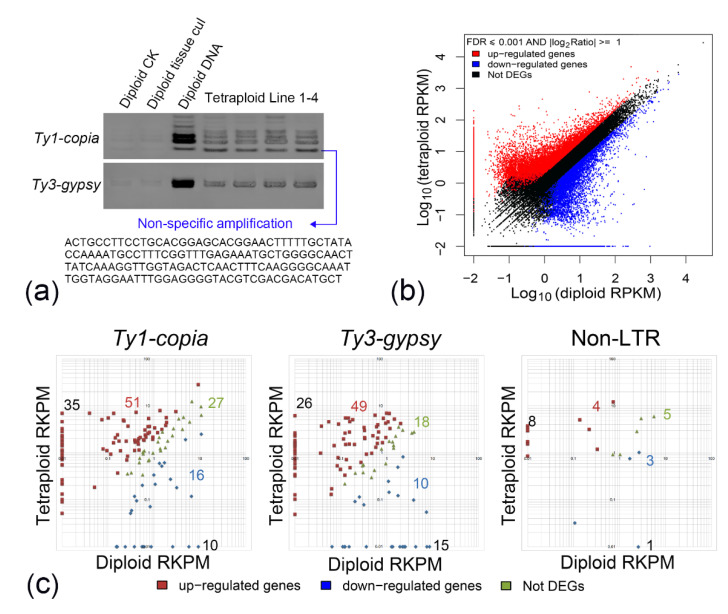
Transcriptional activation of LTR retrotransposons following autopolyploidization. (**a**) Polyacrylamide gel electrophoresis of the Ty1RT and Ty3RT sequences of diploid (tissue-cultured and non-tissue-cultured) and tetraploid species of *C. nankingense*. The Ty1RT bands at the bottom of the gel show non-specific amplification. (**b**) Analysis of the differentially expressed genes of diploid and tetraploid species of *C. nankingense*. (**c**) Analysis of the Ty1-*copia*, Ty3-*gypsy*, and non-LTR-type retrotransposons of diploid and tetraploid species of *C. nankingense*.

**Figure 5 plants-11-00315-f005:**
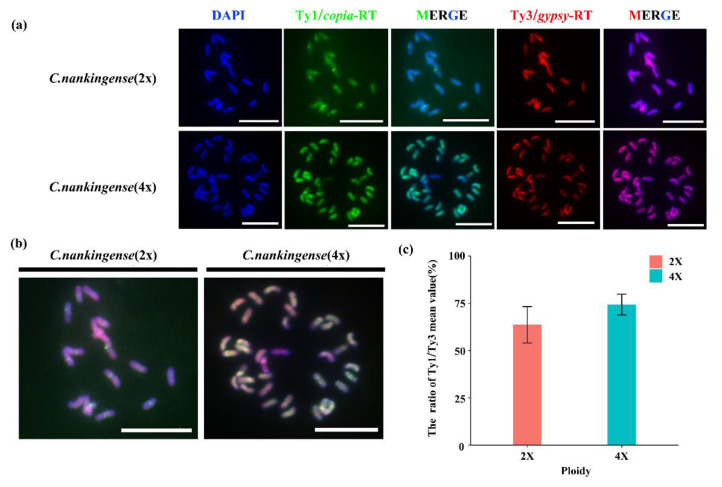
Fluorescence in situ hybridization (FISH) analysis of diploid and tetraploid species of *C. nankingense* (bar = 10 μm). (**a**) FISH result of Ty1/*copia*-RT (green) and Ty3/*gypsy*-RT (red) in diploid and tetraploid species of *C. nankingense*. (**b**) All channels merged images of diploid and tetraploid species of *C. nankingense.* (**c**) The statistical results of fluorescence intensity between diploid and tetraploid species of *C. nankingense*.

**Figure 6 plants-11-00315-f006:**
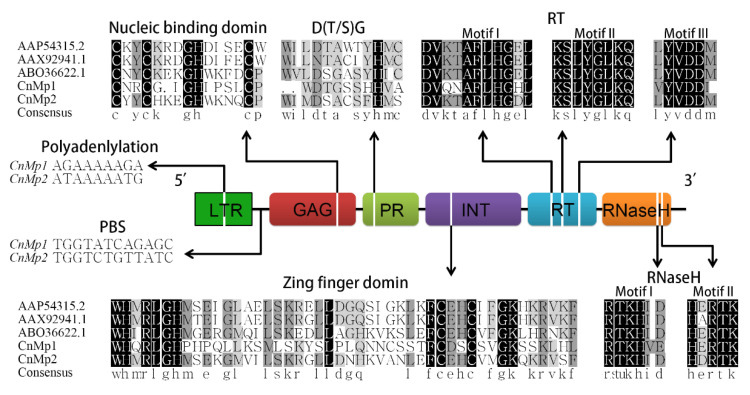
The conservative motifs of CnMp1and CnMp2. Entire conserved signatures among retroelements are indicated with black shading; partially conserved signatures are indicated with grey shading.

**Figure 7 plants-11-00315-f007:**
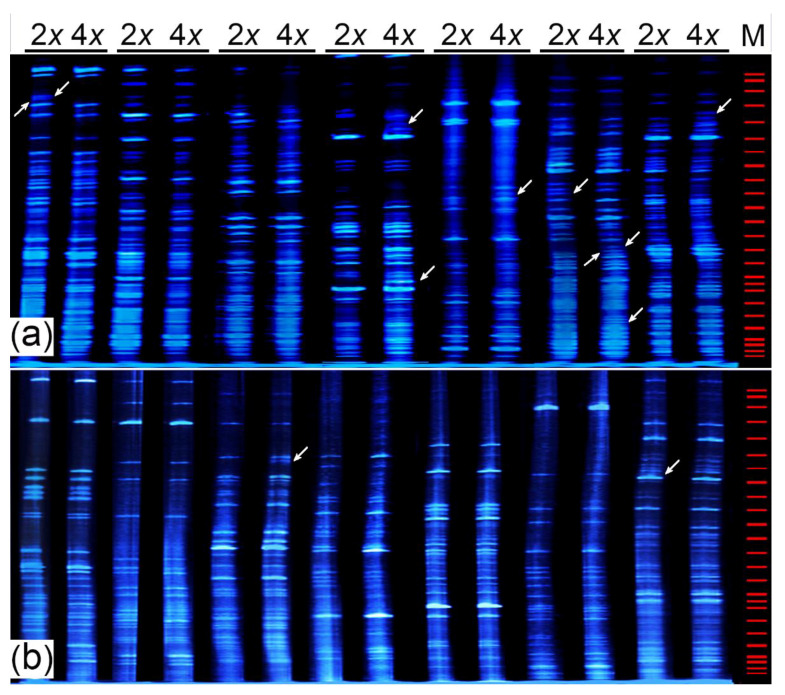
SSAP profiles of diploid and tetraploid (T1) species of *C. nankingense*. The LTR-specific primers were designed from *CnMp1* (**a**) and *CnMp2* (**b**).

**Table 1 plants-11-00315-t001:** Analysis of sequence premature transcription termination and the fragment length of Ty1RT.

Fragment Length	Premature Transcription Termination Number/Total Number	Percentage
263	1/37	2.7%
216	4/4	100%
221	1/1	100%
225	1/1	100%
226	2/2	100%
231	1/1	100%
232	4/4	100%
250	1/1	100%
254	2/2	100%
255	0/1	0%
261	1/1	100%
264	1/1	100%
265	1/1	100%
266	0/7	0%
274	1/1	100%
276	0/1	0%
302	1/1	100%
317	3/3	100%
332	1/1	100%
386	4/4	100%
401	1/1	100%

## Data Availability

Data is contained within the article or Appendix A. The data presented in this study are available in Appendix A here.

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
