# Peer review of "An Eruption of LTR Retrotransposons in the Autopolyploid Genomes of Chrysanthemum nankingense (Asteraceae)"

_plants, 2022, doi:10.3390/plants11030315_

Round 1
Reviewer 1 Report
Polyploidization is an important process in plant evolution. Differ from allopolyploidization, genome shock during auto-polyploidization is far from clear. In this manuscript, He and colleagues described the transposon transcriptional alterations and transposition in 4x Chrysanthemum nankingense. The results are pretty interesting for the society of polyploidization. However, I do have some concerns about the results:
- It is not clear how many generations of the 4x C. nankingense were propagated? How do the authors exclude the possibility that TEs become active in tissue-cultured C. nankingense after the same generations? The correct control is critical to conclude the effects of polyploidization.
- It is not clear how many times the authors repeat the experiments of AFLP analysis. This information is also important to evaluate the results.
Author Response
Dear Editors and Reviewers:
Thank you for your letter and for the reviewers` comments concerning our manuscript entitled “An eruption of LTR retrotransposons in the autopolyploid genomes of Chrysanthemum nankingense (Asteraceae)” (ID: plants-1572967). Those comments are all valuable and very helpful for revising and improving our paper, as well as the important guiding significance to our researches. We have studied comments carefully and have made correction which we hope meet with approval. Revised portion are marked in red in the paper. The main corrections in the paper and the responds to the reviewer`s comments are as flowing:
Reviewer: 1
1) It is not clear how many generations of the 4x C. nankingense were propagated? How do the authors exclude the possibility that TEs become active in tissue-cultured C. nankingense after the same generations? The correct control is critical to conclude the effects of polyploidization.
A:We propagated the 4x C. nankingense by stem cutting containing terminal bud in the 3:1 mixture of garden soil and vermiculite at 25℃ to conclude the minimum effects of polyploidization. Besides, the results showed in the Figure 4a also suggests that TEs inactive in the tissue culture seedlings of diploid C. nankingense.
2) It is not clear how many times the authors repeat the experiments of AFLP analysis. This information is also important to evaluate the results.
A:The number of biological and technical replications were added in the line 590-591.

Reviewer 2 Report
The authors studied to evaluate the genome shock following autopolyploidization by C. nankingense autotetraploids (Asteraceae). According to the data, the genetic modifications entailed the loss of old fragments and the gain of new fragments, with some new sequences being potential LTR retrotransposons. RT-PCR, transcriptome sequencing, and LTR retrotransposon-based molecular marker approaches were used to examine the dynamics of Ty1-copia and Ty3-gypsy. Furthermore, the results of fluorescence in situ hybridization (FISH) suggest that autopolyploidization may be accompanied by LTR retrotransposon perturbations, and that emergence retrotransposon insertions may show more rapid divergence, resulting in diploid-like behavior, potentially speeding up the evolutionary process among progenies.
The author writes the article in an organized way that makes easy to follow it. The importance of the research is a new data about autopolyploidization and behavior of retrotransposons in genome.
I recommend the article for publication.
I suggest the following corrections:
Line 52: Please add explanation for the abbreviation: TEs (Transposable elements)
Line 133: Correct the spaces: plants[40] .
Author Response
Dear Editors and Reviewers:
Thank you for your letter and for the reviewers` comments concerning our manuscript entitled “An eruption of LTR retrotransposons in the autopolyploid genomes of Chrysanthemum nankingense (Asteraceae)” (ID: plants-1572967). Those comments are all valuable and very helpful for revising and improving our paper, as well as the important guiding significance to our researches. We have studied comments carefully and have made correction which we hope meet with approval. Revised portion are marked in red in the paper. The main corrections in the paper and the responds to the reviewer`s comments are as flowing:
Reviewer 2:
1)Line 52: Please add explanation for the abbreviation: TEs (Transposable elements)
A:We have added the abbreviation in the line 52 of the new manuscript.
2)Line 133: Correct the spaces: plants[40] .
A:We have modified the mistake in the line 132 of the new manuscript.

Reviewer 3 Report
Dear editor and colleagues,
I have read with great interest the entitled paper ‘An eruption of LTR retrotransposons in the autopolyploid genomes of Chrysanthemum nankingense (Asteraceae)’ submitted to mdpi-plants
It is a very interesting study focusing on the retrotransposon’s activation/inactivation and study of the following genomic events caused by autopolyploidization in chrysanthemum
According to my opinion, this is a well-focused, designed, executed and written study that completes a series of publication of the group. It combines data at the genomic, transcriptional and structural level using a plethora of appropriate techniques. As a result, this study has merit for publication as it stands.
However, I have a few points/suggestions that the authors could consider.
The scope of the study is not very clear (in the last paragraph of the introduction) and adjustments are needed.
Regarding figure 1. I understand that flow cytometry is not a part of the current study but an internal reference marker plant was not used and different nuclei extractions can have an effect on the flow cytometer histogram. Also, it is not clear why the authors selected the FL4 filter (usually for dapi or PI staining other filters (FL2) are more appropriate)
L171. It is not clear what you mean by ‘recycled’
Since retrotransposons are a significant part of the genome, it would be interested to indicate (if you have data) or comment the affect of loss or gain of specific loci to the overall genomic content (C-values). From my experience, auto or allo-polyploidy can have a significant effect in genome downsizing (since C-values of the chromosomal doubled plant differ significantly from the theoretical doubled values).
Based on the above, my recommendation is a minor revision
Author Response
Dear Editors and Reviewers:
Thank you for your letter and for the reviewers` comments concerning our manuscript entitled “An eruption of LTR retrotransposons in the autopolyploid genomes of Chrysanthemum nankingense (Asteraceae)” (ID: plants-1572967). Those comments are all valuable and very helpful for revising and improving our paper, as well as the important guiding significance to our researches. We have studied comments carefully and have made correction which we hope meet with approval. Revised portion are marked in red in the paper. The main corrections in the paper and the responds to the reviewer`s comments are as flowing:
Reviewer 3:
1) The scope of the study is not very clear (in the last paragraph of the introduction) and adjustments are needed.
A: We have added the clear scope in the line 150-151.
2) Regarding figure 1. I understand that flow cytometry is not a part of the current study but an internal reference marker plant was not used and different nuclei extractions can have an effect on the flow cytometer histogram. Also, it is not clear why the authors selected the FL4 filter (usually for dapi or PI other filters (FL2) are more appropriate)
A: â‘ The information about the internal reference marker plant was C.lavandulifolium (2n=2x=18) and related flow cytometer data added in the figure 1.
â‘¡ We extracted the samples by the same method and kit showed in line 569-575 to ensure there was no significant differences in the studies.
③ We utilized the DPAI staining and acquired the flow cytometer histogram by FL4 filter in order to examine the changes in ploidy,which was coincided with the method by conducting the FL2 filters and using PI staining.
3) L171. It is not clear what you mean by ‘recycled’
A: We have modified the mistake in the line 172.
4) Since retrotransposons are a significant part of the genome, it would be interested to indicate (if you have data) or comment the affect of loss or gain of specific loci to the overall genomic content (C-values). From my experience, auto or allo-polyploidy can have a significant effect in genome downsizing (since C-values of the chromosomal doubled plant differ significantly from the theoretical doubled values).
A: Artemisia species are relative genus of Chrysanthemum genus. In the research entitled” Genome size dynamics in Artemisia L. (Asteraceae): following the track of polyploidy“, the results showed that C-values have a convergent linear relationship with the increase of ploidy. Furthermore, the more obvious the difference of C-values among the higher the ploidy. Besides, our previous works entitled “Rapid genetic and epigenetic alterations under intergeneric genomic shock in newly synthesized Chrysanthemum morifolium x Leucanthemum paludosum hybrids (Asteraceae)“ and “Rapid genomic and transcriptomic alterations induced by wide hybridization: Chrysanthemum nankingense x Tanacetum vulgare and C. crassum x Crossostephium chinense (Asteraceae)” indicated that the fragment elimination occupied major parts after polyploidization.
Meanwhile, the result of this studies also found that the C. nankingense autotetraploid showed 1.6%-3.0% fragment elimination and 0.7%-1.4% novel fragment emergence, which may lead to convergent genome size to a greater extent. Consequently, we added the related discussion in the line 400-405.
